# Congenital Anterior Dislocation of the Sacrococcygeal Bone in a Newborn

**DOI:** 10.3390/diagnostics13122108

**Published:** 2023-06-19

**Authors:** Artur Fabijan, Bartosz Polis, Krzysztof Zakrzewski, Agnieszka Zawadzka-Fabijan, Sara Korabiewska-Pluta, Emilia Nowosławska

**Affiliations:** 1Department of Neurosurgery, Polish-Mother’s Memorial Hospital Research Institute, 93-338 Lodz, Poland; jezza@post.pl (B.P.); krzysztof.zakrzewski@iczmp.edu.pl (K.Z.); emilia.nowoslawska@iczmp.edu.pl (E.N.); 2Department of Rehabilitation Medicine, Medical University of Lodz, 90-419 Lodz, Poland; agnieszka.zawadzka@umed.lodz.pl; 3Department of Cardiology, Polish-Mother’s Memorial Hospital Research Institute, 93-338 Lodz, Poland; sara.korabiewska@gmail.com

**Keywords:** congenital anomalies, myelomeningocele, lumbosacrococcygeal agenesis, caudal regression syndrome

## Abstract

We present a case of a child who was transported to the Neurosurgery Clinic from another hospital for the purpose of performing a surgical procedure of the spinal myelomeningocele. On the first day of the stay, a set of tests was performed, including an anterior-posterior (AP) projection X-ray, which clearly showed a developmental defect in the lumbar-sacral section of the spine. In the follow-up physical examination, there was a depression of the skin on the right side of the surgical scar after closing the open myelomeningocele. In the follow-up MRI of the lumbar-sacral section, an extremely rare congenital anterior dislocation of the sacrococcygeal bone was unexpectedly visualized. Despite recommendations for further diagnostics, the patient did not attend the required follow-up examinations. In the final section, we provide a general summary of the literature on rare developmental defects of the spine in children.

A female newborn was delivered via cesarean section (due to threatening fetal distress) with a diagnosed lumbar myelomeningocele and significant lower limb deformities after birth. The child was transported to our institute for further treatment. According to the medical record, the initial prenatal ultrasound was normal. In the 30th week of pregnancy, intrauterine growth restriction (IUGR) was diagnosed, and on the day of delivery, oligohydramnios and ventricular system enlargement of the central nervous system (CNS) were detected. On the first day of the patient’s stay, a set of tests was performed, including an anterior-posterior (AP) projection X-ray. The study shows a developmental defect of the lumbar-sacral spine (Figure 1).

The child was qualified for a hernia repair surgery. A medical rehabilitation specialist diagnosed the lower limbs positioned in flexion and adduction at the hip joints (trace movement), extended in the knee joints (trace flexion—more frequently in hyperextension), and feet in a stable equinovarus position. The patient’s CT scan confirmed ventriculomegaly without evident signs of active hydrocephalus. Ultrasound examination did not reveal any residual urine in the bladder after voiding. In the follow-up physical examination, there was a depression of the skin on the right side of the surgical scar after closing the open myelomeningocele. An MRI examination was performed to exclude the presence of a dermal sinus, which revealed a developmental defect of the sacrococcygeal bone (Figure 2). Despite recommendations for further diagnostics, the patient did not attend the required follow-up examinations.

In this study, we present what we believe to be an extremely rare developmental defect of the lumbar-sacral spine. Although the parents did not continue the therapeutic procedure despite the recommendations, further stages such as control medical visits for the purpose of assessing the clinical neurological state and development of the child are crucial in such complex pathologies. In addition, control radiological examinations, MRI, X-ray, CT, or others, depending on the clinical and neurological condition of the patient, would allow for an assessment of the behavior and dynamics of the defect. We must not forget that the child underwent surgical treatment for a spinal myelomeningocele. Control tests for the evaluation of the structures of the spinal canal and CNS are essential. Surgical procedures are exceptionally difficult to assess, due to the uniqueness of the defect, and further multi-specialist care is highly recommended.

Congenital defects of the sacrococcygeal segment in infants pose a significant medical challenge and are often diagnosed at the prenatal stage thanks to imaging techniques such as ultrasonography [1]. Various types of these defects include, for instance, lumbosacrococcygeal agenesis or caudal regression syndrome [2,3]. It is particularly important to detect these defects at the earliest possible stage, which enables the initiation of appropriate treatment as soon as possible.

Lumbosacrococcygeal agenesis is a rare developmental defect in which part or all of the lumbar, sacral, and coccygeal vertebrae are missing. This defect is usually associated with multiple other internal organ defects and syndromes, such as Currarino syndrome. Treatment is complex and typically requires a multidisciplinary approach [4].

Caudal regression syndrome is a rare developmental defect characterized by the incomplete development of the lower part of the spine. This defect can affect the function of pelvic organs, including the bladder and intestines, and it can also lead to varying degrees of dysfunction and deformation of the lower limbs. It is often associated with maternal diabetes [5].

Anterior angulation of the coccyx is an anomaly in which the coccyx bone is bent or deviated forward. This can lead to discomfort and pain, especially when sitting. Causes of this condition may include trauma, childbirth, or congenital anomalies. Treatment is usually conservative, although surgery may be considered in extreme cases [6].

Myelomeningocele is another serious defect that involves the presence of an open hernia in the spinal area, through which the meninges and spinal cord are visible. This results in damage to the nervous system, leading to various neurological complications, such as paralysis of the lower limbs [7].

An important aspect of treating these defects is a comprehensive approach that includes both medical and psychological aspects. Although advancements in research on congenital sacrococcygeal segment defects are promising, there are still many challenges. Preventing sacrococcygeal segment developmental defects is also an area of active research. Many of these defects may be associated with environmental factors, such as exposure to certain chemicals during pregnancy. Folic acid supplementation is one proven way to prevent many spinal developmental defects [8].

In summary, congenital defects of the sacrococcygeal segment are a serious medical challenge that requires an integrated approach that includes early defect detection, effective treatment methods, and support for families. Achievements in research on these defects are promising, but there is still much to understand in order to provide better outcomes for patients affected by these conditions.

## Figures and Tables

**Figure 1 diagnostics-13-02108-f001:**
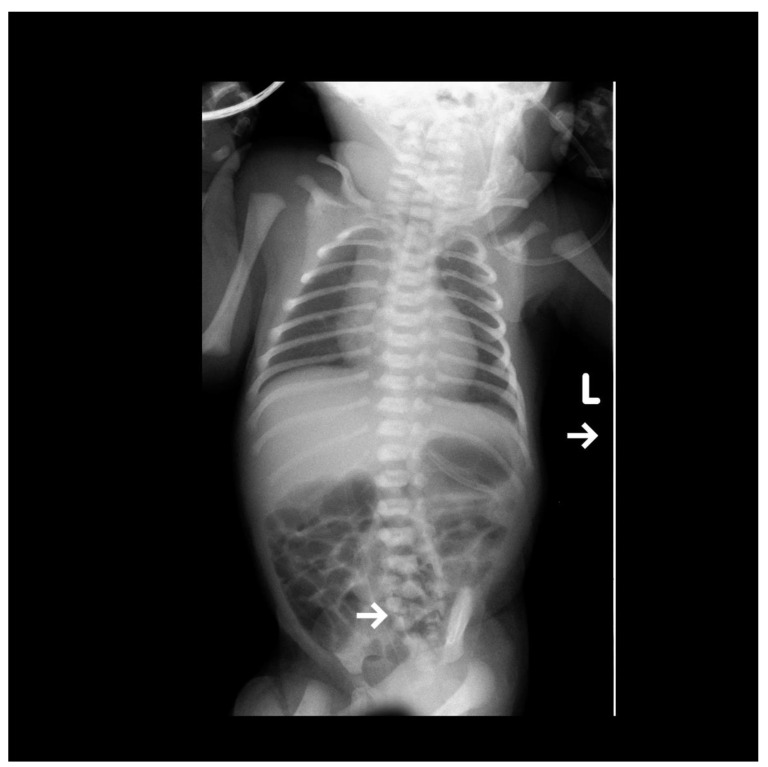
The visible AP projection X-ray shows bilateral small parenchymal consolidations in the upper lung fields. The vascular catheter ends at the level of Th11, projecting onto the liver. The end of the gastric probe is seen projecting onto the organ. Additionally, a clear developmental defect of the lumbar-sacral spine is visible, indicated by an arrow.

**Figure 2 diagnostics-13-02108-f002:**
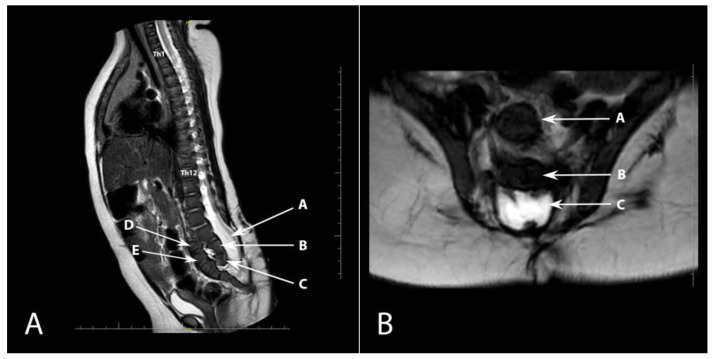
(**A**). Sagittal plane MRI T2-weighted sequence showing anterior dislocation of the sacrococcygeal bone (arrow). Th1 and Th12 vertebrae are also marked. A—open spinal canal, B—L5 vertebra, C—hypoplastic S1 vertebra, D—hypoplastic S2 vertebra, E—S3 vertebra; (**B**). Axial plane T2-weighted MRI sequence. A—S3 vertebra, B—L5 vertebra, C—open spinal canal.

## Data Availability

Not applicable.

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
