# Peer review of "Congenital Anterior Dislocation of the Sacrococcygeal Bone in a Newborn"

_diagnostics, 2023, doi:10.3390/diagnostics13122108_

Round 1
Reviewer 1 Report
This manuscript aims to present an extremely rare congenital anterior dislocation of the sacrococcygeal bone in a Newborn using MRI as the Interesting Image. The topic can be interesting for the readers; the study is well-designed and the manuscript is well-written. I recommend accepting it.
This manuscript aims to present an extremely rare congenital anterior dislocation of the sacrococcygeal bone in a Newborn using MRI as the Interesting Images. The topic can be interesting for the readers; the study is well-designed and the manuscript is well-written. I recommend accepting it.
Author Response
We are extremely grateful for your positive review
Sincerely,
Artur Fabijan
Reviewer 2 Report
Please clarify if An X-ray (even AP) was obtained in this patient. If available include the radiographs as an additional image.
in the abstract: start the section describing the case, and end the section declaring that you are going to summerize the literature.
in the main section, please underline the importance of imaging for diagnosis and treatment planning. if X-ray is available comment the role of radiograph or CT in comparison to MRI.
Treatment was not performed, but some hypothetic approaches could be described for the case presented, for example checking from the literature for similar problems.
Also, if possible, give an overview on the possible complications of an untreated case, and the eventual need for imaging follow-up
